# CAReNet : a novel architecture for low data regime mixing convolutions and attention

## Abstract

In the rapidly evolving landscape of deep learning for computer vision, various architectures have been proposed to achieve state-of-the-art performance in tasks such as object recognition, image segmentation, and classification. While pretrained models on large datasets like ImageNet have been the cornerstone for transfer learning in many applications, this paper introduces **CAReNet** (**C**onvolutional **A**ttention **Re**sidual **Net**work), a novel architecture that was trained from scratch, in the absence of available pretrained weights. CAReNet incorporates a unique blend of convolutional layers, attention mechanisms, and residual connections to offer a holistic approach to feature extraction and representation learning. Notably, CAReNet closely follows the performance of ResNet50 on the same training set while utilizing fewer parameters. Training CAReNet from scratch proved to be necessary, particularly due to architectural differences that render feature representations incompatible with those from pretrained models. Furthermore, we highlight that training new models on large, general-purpose databases to obtain pretrained weights requires time, accurate labels, and powerful machines, which causes significant barriers in many domains. Therefore, the absence of pretrained weights for CAReNet is not only a constraint but also an opportunity for architecture-specific optimization. We also emphasize that in certain domains, such as space and medical fields, the features learned from ImageNet images are vastly different and can introduce bias during training, given the gap that exists between the domains of pretraining and the task of transfer learning. This work focuses on the importance of architecture-specific training strategies for optimizing performance and also demonstrates the efficacy of CAReNet in achieving competitive results with a more compact model architecture. Experiments were carried out on several benchmark datasets, including Tiny ImageNet, for image classification tasks. Signifying a groundbreaking stride in efficiency and performance, CAReNet not only outpaces ResNet50 by achieving a lead of 2.61% on Tiny-Imagenet and 1.9% on STL10, but it does so with a model that's nearly half the size of ResNet50. This impressive balance between compactness and elevated accuracy highlights the prowess of CAReNet in the realm of deep learning architectures.

## 1 Introduction

In the landscape of machine learning and artificial intelligence, supervised learning techniques have demonstrated remarkable success in solving a multitude of complex tasks, ranging from computer vision to natural language processing. Despite their efficiency, these approaches come with a set of inherent limitations that cause significant challenges for broad adoption. Foremost among these is the insatiable appetite for labeled data, which not only necessitates considerable human effort but often requires domain expertise for accurate annotation. The need for vast amounts of training data creates bottlenecks in data acquisition and preparation, limiting the applicability of supervised methods in low-resource settings or specialized domains where labeled examples are scarce. Even when an adequate dataset is assembled, the computational cost of training sophisticated models is another critical concern. The required hardware for high-quality model training, typically comprising multiple GPUs or TPUs, presents a substantial financial burden. Furthermore, the time investment for model training and hyperparameter tuning is significant, making it a resource-intensive endeavor

that could span days or even weeks for particularly complex models. Thus, while supervised learning methods continue to push the envelope in predictive performance, the trade-offs in terms of data, computational resources, and time raise important questions about their practicality and scalability in real-world applications.

It is within this challenging framework that we propose a novel architecture designed to mitigate these inherent difficulties. Uniquely, our model is not pre-trained on a large, external dataset but is instead tailored to perform exceptionally well using only the dataset in focus. By eschewing the need for expansive labeled datasets and high-computational training environments, our approach aims to democratize the application of machine learning models. This not only reduces the time and financial costs associated with training but also enhances the model's applicability in specialized or low-resource settings. In doing so, we present a feasible alternative to the traditional paradigms of supervised learning, emphasizing efficiency and accessibility without compromising on performance.

In the domain of image classification, Convolutional Neural Networks (CNNs) have long been considered the gold standard, achieving state-of-the-art results in a variety of benchmarks Smith & Topin (2016); Johnson & Zhang (2017). Their hierarchical structure naturally lends itself to the progressive abstraction of features, from edges to complex shapes, making them exceptionally well-suited for tasks that require spatial awareness Lecun et al. (1998). However, the recent surge of Vision Transformers (ViTs) has opened new avenues for exploration Dosovitskiy et al. (2020). These architectures leverage self-attention mechanisms to capture long-range dependencies and have shown comparable, if not superior, performance to CNNs on several classification benchmarks Wang et al. (2021). However, ViTs often require large amounts of data and computational resources for effective training, making them less accessible for low-resource settings Brown et al. (2020). Despite these limitations, the flexibility of Vision Transformers in handling different modalities and their capability for better interpretability make them an attractive alternative to traditional CNNs Vaswani et al. (2017).

Despite their data-intensive nature, ViTs have shown remarkable performance even in low-data regimes when equipped with appropriate regularization and transfer learning techniques Parvaiz et al. (2023). Their ability to capture both local and global features makes them a compelling choice for image classification tasks, often outperforming state-of-the-art CNNs under similar conditions.

The fusion of Convolutional Neural Networks (CNNs) and Vision Transformers (ViTs) Khan et al. (2022) presents a groundbreaking approach to image classification, capitalizing on the unique strengths of each architecture. On the one hand, CNNs excel in parameter efficiency and local feature extraction, making them well-suited for tasks that require a nuanced understanding of spatial hierarchies. On the other hand, ViTs are adept at capturing global contextual information through self-attention mechanisms, providing a more holistic understanding of the image. By integrating these two architectures, a hybrid model can leverage the local feature recognition of CNNs and the global contextual awareness of ViTs, creating a more robust and versatile system for image classification.

The remainder of this paper is organized as follows: The section 2 provides a comprehensive review of existing literature and precedents in the domain. In the section 3, we elaborate on our proposed architectural design, discussing the datasets employed and elaborating on the specific training parameters. Our experiments' outcomes and findings are showcased in the section 4. The paper concludes with a summary of our contributions, their potential implications, and future perspectives.

## 2  RELATED WORKS

In the area of computer vision, Convolutional Neural Networks have long been the cornerstone for tackling a myriad of tasks, ranging from simple image classification to complex object detection. However, the recent emergence of Vision Transformers has ushered in a new era, challenging the traditional paradigms and offering fresh perspectives on how vision problems can be approached. While these Transformer-based models have shown remarkable promise, they are not without their challenges, particularly when it comes to computational efficiency and the ability to scale. Various innovative solutions have been proposed to mitigate these issues, such as introducing sparse attention mechanisms and developing hybrid models that marry the best of both worlds—ConvNets and

Transformers. In this context, authors in Dai et al. (2021), proposed a novel hybrid architecture, CoAtNet, that combines the strengths of Convolutional Neural Networks (CNNs) and Transformers. While Transformers excel in terms of model capacity, they suffer from poor generalization due to the absence of the right inductive bias. On the other hand, CNNs possess this inductive bias but may lack the model capacity for certain tasks.

In Dagli (2023), authors introduced a novel method designed for learning from limited data to classify galaxy morphologies. The approach employs a hybrid architecture that combines Transformer and Convolutional Neural Networks (CNNs). To enhance the model's performance, data augmentation and regularization techniques are also incorporated. They achieves state-of-the-art results in predicting galaxy morphologies and demonstrates strong performance on other datasets like CIFAR-100 and Tiny ImageNet. The paper also delves into the challenges of using large models and training methods in low-data regimes. It explores semi-supervised learning approaches that utilize both labeled and unlabeled data from different datasets. Despite promising results reported for the Astroformer architecture, the absence of publicly available code prevented us from conducting our own experiments to challenge these findings.

Authors in the paper Liu et al. (2023) addressed the computational inefficiencies of Vision Transformers (ViTs) by proposing a new family of models called EfficientViT. They identify memory access, computation redundancy, and parameter usage as the three main bottlenecks affecting the speed of ViTs. They introduce a new building block with a "sandwich layout," which places a single memory-bound Multi-Head Self-Attention (MHSA) layer between efficient Feed-Forward Network (FFN) layers. This design improves memory efficiency and enhances channel communication. The paper also introduces a Cascaded Group Attention (CGA) module that feeds different splits of the full feature to different attention heads, reducing computational redundancy and improving attention diversity. EfficientViT models outperform existing efficient models in terms of both speed and accuracy. For instance, EfficientViT-M5 surpasses MobileNetV3-Large by 1.9% in accuracy while achieving higher throughput on both GPU and CPU.

In Tu et al. (2022), authors presented a new type of Transformer module, called multi-axis self-attention (MaxVit), that capably serves as a basic architecture component which can perform both local and global spatial interactions in a single block. Compared to full self-attention, MaxVit enjoys greater flexibility and efficiency, i.e., naturally adaptive to different input lengths with linear complexity; in contrast to (shifted) window/local attention, MaxVit allows for stronger model capacity by proposing a global receptive field.

In Barhoumi & Ghulam (2021), authors proposed a hybrid architecture that leverages the strengths of both Convolutional Neural Networks and Vision Transformers. By utilizing multiple CNNs for feature extraction, they enrich the input that the Vision Transformer attends to, thereby enhancing its ability to capture relevant features. This approach not only improves the performance of the Vision Transformer but also introduces a degree of modularity and scalability, making it adaptable for various medical imaging tasks.

All the previous recent works focused on fusion ViT and CNN. Our approach aligns with the same paradigm, albeit with an emphasis on more effectively, flexibility and low data dependency integrating the two components. We will elaborate on our architecture in the subsequent section.

## 3 METHODOLOGY

### 3.1 THE PROPOSED ARCHITECTURE

As depicted in Figure 1, our architecture starts with an introductory convolutional block, tailored to extract elementary feature maps from the input image. Subsequent to this, we introduce a duo of parallelized, distinct blocks. The first block within this duo integrates a bottleneck block, composed of multiple convolutional layers, offering enhanced feature extraction capabilities. Simultaneously, the secondary block introduces a residual unit, explicitly designed to bolster information flow and empower the model to learn profounder representations. The resultant feature maps from these parallel constructs are seamlessly merged using an element-wise addition operation, which is then succeeded by a Max pooling layer to further refine the spatial hierarchy of the feature maps.

This modular assembly is systematically reiterated thrice, employing varying channel dimensions to adeptly capture multi-scale feature intricacies. The terminal segment of our architecture encompasses a CareNet block, which is structured with a bottleneck block followed by an attention mechanism operating in parallel across both grid and window patterns. The outputs from these attention mechanisms are subsequently averaged to obtain a consolidated feature representation. Running in tandem with the CareNet block is an additional residual unit, ensuring consistent feature propagation.

To conclude the architectural design, we employ a Max pooling layer, which is then followed by a flatten operation to reshape the feature maps into a singular vector. The final component of our architecture is a linear classification layer, dedicated to producing the predictive outcomes. In the following sections, we delineate the advantages of the various layers employed in our architecture.

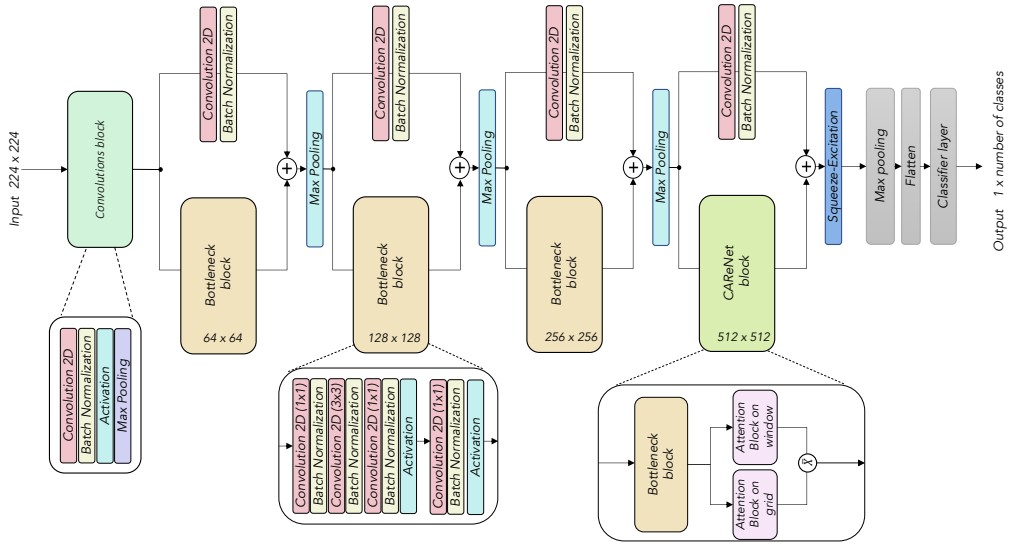

Figure 1: Illustration of our architecture

**Convolutional Layers**   Convolutional layers serve as the building blocks for many state-of-the-art computer vision architectures, including VGG presented in Simonyan & Zisserman (2015) and ResNet in He et al. (2015). These layers are particularly effective in capturing local features like edges, corners, and textures in an image and preserving the spatial relationships between pixels. The virtues of Convolutional Neural Networks (CNNs) are manifold. Initially, their architecture is inherently well-suited for capturing hierarchical patterns in the data, enabling them to excel particularly in tasks that necessitate an understanding of spatial hierarchies, such as object recognition and segmentation. Subsequently, their parameter-sharing mechanism enhances computational efficiency, thereby permitting the training of deeper, more complex models without a commensurate increase in computational resources.

**Max Pooling**   Max-pooling layers have become an integral component in numerous convolutional neural network architectures due to their numerous advantages. Primarily, max-pooling reduces the spatial dimensions of the feature maps, leading to a decrease in the amount of parameters and computations and thus avoid the overfitting in the network. This reduction helps in combating the curse of dimensionality, enabling the network to focus on the most salient features and thus enhancing its generalization capability. Notably, max-pooling layers also play a pivotal role in hierarchically capturing abstract representations in deep CNNs. In sum, the inclusion of max-pooling layers in CNN architectures offers a systematic approach to achieving computational efficiency, robustness, and a hierarchical representation of features, making them indispensable in modern deep learning paradigms.

**Attention Mechanisms**   Attention mechanisms have gained prominence primarily in the field of natural language processing with architectures like Transformers presented in Vaswani et al. (2017). However, their application in computer vision, as seen in models like ViT in Dosovitskiy et al. (2020), has shown significant promise. Attention layers weigh the importance of different regions in the image, allowing the model to focus on more relevant parts. They capture long-range dependencies without requiring higher computational costs.

**Squeeze-and-Excitation Layers**   Authors in Hu et al. (2018) presents the Squeeze-and-Excitation (SE) layers which offer a mechanism to adaptively recalibrate channel-wise feature responses. SE layers dynamically adjust the importance of each channel based on the global average pooling. These layers add minimal computational overhead while offering performance gains. Finally, by allowing the model to focus on more informative channels, SE layers improve the representational power of the network.

**Residual Connections**   Introduced by He et al. in ResNet He et al. (2015), residual connections have revolutionized the training of deep networks. In our architecture, Squeeze-and-Excitation (SE) layers are incorporated for a variety of reasons, notably to enhance the network's capacity for feature recalibration, thereby leading to a more adaptive representation of input data. Additionally, SE layers contribute to the overall interpretability of the model by providing a mechanism to weigh the importance of different features.

### 3.2 EXPERIMENTATIONS

**Methodology**   In the methodology adopted for this study, we utilize several benchmark datasets, each of which is elaborated upon in the *Data* section. To ensure a fair and unbiased comparison, we commit to training a variety of models from scratch, or "from zero", without the influence of any pre-existing weights or pre-trained parameters. This approach ensures that each model's performance is a direct result of its architecture and training regime, devoid of any external influences. All experiments conducted herein are grounded in our original efforts, employing the bare codebases of the models without any pre-loaded weights, ensuring the purity and repeatability of our results.

**Data**   In this study, we leverage a diverse set of benchmark datasets to rigorously evaluate the performance and robustness of our proposed models. The Tiny-ImageNet dataset Le & Yang (2015), a downscaled variant of the much larger ImageNet Deng et al. (2009), comprises 200 classes with each class having 500 training and 50 validation images of 64x64 resolution. It provides a challenging yet computationally accessible platform for fine-grained classification tasks. For the Tiny-ImageNet dataset, we opted for a reduced set of 450 training images per class, rather than the full 500, in an effort to investigate the model's performance under a more constrained training scenario. Additionally, we used 50 images from each class for validation to evaluate the generalization capabilities of our approach.

We also utilize the Fashion-MNIST dataset Xiao et al. (2017), an alternative to the original MNIST dataset LeCun & Cortes (2005) of handwritten digits. Fashion-MNIST contains grayscale images of 10 different clothing types, such as shirts, trousers, and footwear, each normalized to a 28x28 pixel resolution. This offers a more intricate classification landscape compared to MNIST, which features single-channel, 28x28 grayscale images of numerals 0-9 and serves as a cornerstone for basic machine learning algorithms.

Finally, we incorporate the STL-10 dataset Coates et al. (2011), designed for unsupervised and transfer learning. STL-10 contains 10 classes of 96x96 color images, with 500 labeled training instances and 800 test instances per class, in addition to an unlabeled set of 100,000 images. The selection of these datasets provides a broad testing ground to validate the versatility and scalability of our methods across different domains and levels of complexity.

**Training parameters**   During the training phase, we used an RTX 3090 graphics card and opted for a maximum of 100 epochs to ensure comprehensive model training. Nevertheless, understanding the risk of overfitting and to capitalize on computational efficiency, we employed an early stopping mechanism. This mechanism monitors the validation loss during training. If the validation loss fails to improve over a span of 10 consecutive epochs, referred to as 'patience', the early stopping pro-

| Architecture | Nbr of params (Million) | Model size (MB) | Top1 Validation Accuracy (%) | | | |
|---|---|---|---|---|---|---|
| | | | STL10 | Fashion Mnist | Mnist | Tiny-Imagenet (ours) |
| ResNet50 | 23.53 | 91 | 78.36 | 95.15 | 99.35 | 51.79 |
| ResNet34 | 21.28 | 82 | 76.72 | 95.11 | 99.43 | 56.19 |
| ResNet18 | 11.18 | 44 | 75.33 | 94.37 | 99.35 | 54.35 |
| CoatNet_0 | 17.02 | 67 | 74.75 | 94.99 | 99.17 | 54.56 |
| MaxVit | 24.67 | 95 | 76.95 | 93.48 | 99.42 | 58.28 |
| VGG16 | 134.30 | 513 | 10.00 | 94.91 | 99.33 | 52.57 |
| **CAReNet (ours)** | **12.74** | **50** | **80.26** | **94.94** | **99.4** | **54.4** |

Table 1: Top-1 accuracy and model size comparison of neural network architectures trained from scratch across multiple datasets

cedure is triggered, and the training is halted. This approach not only aids in preventing overfitting but also ensures that the model training is efficient, by eliminating the redundancy of epochs that do not contribute to model improvement. In our experimentation phase, meticulous attention was given to the selection of hyperparameters to optimize model performance across various image datasets. A learning rate of $1 \times 10^{-4}$ was universally applied to all datasets, with the notable exception of STL10, for which a learning rate of $5 \times 10^{-4}$ was chosen. These parameters were not arbitrarily set; instead, they emerged as optimal from a series of rigorous experiments. Similarly, the optimizer's selection was grounded in empirical evaluations. After juxtaposing several candidates, the AdamW optimizer was identified as the most efficacious for our task and subsequently adopted.

## 4 RESULTS

In this section, we present a comprehensive evaluation of our proposed architecture, CAReNet, across multiple benchmark datasets. The primary focus of this evaluation is on the Top-1 validation accuracy and the model size, as detailed in Table 1. This comparison is crucial for illustrating the efficiency and effectiveness of CAReNet, particularly when trained from scratch.

### 4.1 PERFORMANCE ACROSS DATASETS

The datasets used for this evaluation include STL10, Tiny ImageNet, Fashion-MNIST, and MNIST. These datasets were chosen to demonstrate the architecture's versatility and performance in diverse scenarios.

#### 4.1.1 STL10 DATASET

For the STL10 dataset, CAReNet achieved a Top-1 accuracy of 80.26%, outperforming ResNet50, which had an accuracy of 78.36%. This was accomplished with nearly half the number of parameters used in ResNet50, demonstrating CAReNet's efficiency in parameter utilization and its effectiveness in balancing accuracy with model compactness.

#### 4.1.2 TINY IMAGENET

On the Tiny ImageNet dataset, CAReNet recorded a Top-1 accuracy of 54.4%. While this is slightly lower than MaxVit's 58.28% accuracy, it is important to note that CAReNet achieved this with a significantly smaller model size, highlighting its capability to handle complex, high-resolution image data effectively.

#### 4.1.3 FASHION-MNIST AND MNIST

CAReNet demonstrates robust performance on Fashion-MNIST and MNIST, with accuracies nearing 95% and over 99%, respectively. These results are indicative of the model's adaptability to different types of image data and complexities, further emphasizing its practical utility.

## 4.2 Model Size and Efficiency

One of the key strengths of CAReNet is its compactness combined with high performance. The architecture's ability to maintain high accuracy with a reduced parameter count and smaller model size makes it a promising candidate for applications where computational resources are limited, such as mobile and edge computing.

## 4.3 Comparative Analysis

The evaluation also included a comparative analysis with other well-established models like ResNet50, ResNet18, VGG16, and emerging architectures like MaxVit and CoatNet. The results show that CAReNet is either equivalent or superior in terms of Top-1 accuracy across the evaluated datasets, while maintaining a smaller model size and fewer parameters.

## 4.4 Implications

The findings from our experiments suggest that CAReNet is not only an efficient and effective architecture for image classification tasks but also a viable solution in scenarios demanding computational frugality. Its ability to deliver high accuracy with a compact model size is pivotal for extending the applicability of deep learning models to a broader range of real-world scenarios, particularly where computational and data resources are scarce.

The presented results furnish an in-depth comparison among several cutting-edge neural network architectures, focusing on their Top-1 validation accuracy across diverse datasets while highlighting their model sizes and parameter counts of the architectures evaluated, ResNet50, with its 23,528,522 parameters, led the pack in the Mnist dataset with an accuracy of 99.35%. However, its performance on STL10, at 78.36%, though commendable, is surpassed by our proposed CAReNet, which achieves an impressive 80.26% while only utilizing 12,745,930 parameters. This is a testament to CAReNet's optimization, effectively leveraging a reduced parameter set to outperform a renowned architecture like ResNet50 on the STL10 dataset.

In summary, the deep learning landscape often poses the challenge of balancing model size, computational efficiency, and performance. However, CAReNet emerges as a sterling example, demonstrating that with judicious architecture design and parameter optimization, it's possible to craft models that are compact yet rival, if not surpass, the performance of their larger counterparts. Our findings model CAReNet's potential as a frontrunner for scenarios demanding computational frugality without compromising on performance.

## 5 Conclusion

In conclusion, the proposed CAReNet architecture shows promise in achieving competitive, if not superior, performance with a relatively lower parameter count, making it an efficient and robust model choice. On the one hand, our CAReNet model showed an improvement in top-1 accuracy, being up to 5.51% higher compared to other architectures. On the other hand, in terms of model complexity, our model requires fewer parameters with a factor ranging from 12.3% more to 46.04% less compared to other architectures. Its consistent performance across various datasets indicates its potential for broad applications in diverse computer vision tasks. Further studies and refinements on this approach could lead to even more efficient and high-performing models suitable for real-world deployment.

As we look toward the future of this research avenue, one of the most compelling next steps involves adapting our CAReNet architecture for semi-supervised and self-supervised learning paradigms. The ultimate goal is to push the envelope further in terms of achieving state-of-the-art performance with minimal labeled data. By leveraging unlabeled data, which is abundantly available but significantly underutilized, we aim to explore the efficacy of CAReNet in settings where acquiring labeled data is either too expensive or impractical. The challenge lies in fine-tuning the architecture and loss functions in a way that maximizes the utility of both labeled and unlabeled data, thereby mitigating the data scarcity issue that is often a bottleneck in deploying machine learning models in real-world applications.

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
