# OpenReview forum: "CARENET : A NOVEL ARCHITECTURE FOR LOW DATA REGIME MIXING CONVOLUTIONS AND ATTENTION"
_ICLR.cc/2024/Conference — Submitted to ICLR 2024_

### Official Review · Reviewer_YEgK · 2023-10-16

**Soundness:** 2 fair
**Presentation:** 2 fair
**Contribution:** 1 poor
**Rating:** 1
**Confidence:** 5

**Summary:**

The authors proposes an architecture for computer vision: a self-attention layer on top of a convolutional network. They claim this is novel, and evaluate on MNIST, Fashion MNIST, STL-10 and Tiny-ImageNet. Their network performs roughly on par with a typical ResNet (better on STL, worse on all other datasets), but has less parameters than a ResNet 50.

**Strengths:**

**Originality**: Putting a Self-Attention Layer on top of a CNN is not novel, and has been proposed many times before (e.g. even in the original Vision Transformer )

**Quality**: The empirical work is poor: Previous work on the same topic is not cited, some prior work is mis-attributed. While the work claims to be targeted towards low-data regimes, it compares only to baselines known to be data-hungry. It does not compare to comparable architectures. There are no error bars even for small scale experiments.

**Clarity:** The work leaves open many questions: the manuscript not explain deviations from standard designs, and talks more about datasets that are well known in the community instead of architectural decisions. Their main contribution is described in a single line short sentence (see "Weaknesses" below).  Numbers in the results are misrepresented to be significant even though they are actually worse.

**Significance:** As far as can be seen from the empirical results, the work is of minor significance.

**Weaknesses:**

Novelty: Variants of this architecture have been proposed many times over. See e.g. the "Related Work" section in the Vision Transformer paper by Dosovitskiy et al. 2021 for some links

Misleading presentations: Table 1 has all of the results for the proposed method boldened. This is usually done to indicate significantly better results, or at the very least BETTER results as comparable methods. The method proposed here is oftentimes performing worse than the competitors, and the use of boldface is misleading, as it is not in line with the norms of the community.

Clarity: The architectural decisions are extremely unclear. No time is spent to give intutions, or even ablations for the concrete choices made. For example:
* The bottleneck blocks used in this work seem fairly similar to ResNet Bottlenet blocks, but have an additional Convolution at the end. No explanation is given for this, it would be nice to have one.
* CAReNeT attention is merely described as "an attention mechanism operating in parallel across both grid and window patterns". I do not understand what that means -- What is a window pattern, what does it mean to be parallel across grids and windows? This is the paper's main contribution, so a lot more explanation should be devoted to it.
* The "Residual Blocks" in the Figure 1 seem to always be downsampling. Why is that? It would be nice if Figure 1 could also point out intermediate resolutions to better understand when & by how much the resolution is reduced.

Significance: Variations of this model were proposed many times before. It is unclear if there is anything special about this version. The experiments do not compare to similar architectures from the literature (again: see the ViT paper for references), so it's unclear if it performs better than those. Table 1 show nice results on STL-10, but the paper does not investigate why this is. Why does CAReNeT perform so well here? Is this due to attention, or due to something else in the architecture? What other applications could benefit from this?

Smaller remarks:

"The fusion of Convolutional Neural Networks (CNNs) and Vision Transformers (ViTs) Khan et al. (2022) presents a groundbreaking approach to image classification"   ==> Khan et al., did not introduce ViT's, nor the fusion of CNNs and ViTs. Both where done by A. Dosovijtsky et al., (2021) (or references therein), please correct this citation.

**Questions:**

I have no questions

---

### Official Review · Reviewer_qugU · 2023-10-30

**Soundness:** 2 fair
**Presentation:** 2 fair
**Contribution:** 2 fair
**Rating:** 3
**Confidence:** 4

**Summary:**

This paper introduces a new network design named CAReNet that incorporates convolutional layers, attention mechanisms,, and residual connections. They focus on network-specific training strategies and the conducted experiment results show the relative improvements to some modern network designs.

**Strengths:**

1. The authors carefully review the different elements of the proposed network and reassemble them into a more compact network.
2. The experiments on STL10, Mnist, Tiny-ImageNet demonstrates the effectiveness and its efficiency in terms of parameters.

**Weaknesses:**

1.Limited novelty. While the paper brings forward a new model design, its foundation lies in leveraging existing techniques. As a result, its contribution is incremental and not enough for establishing a distinct and well-designed model architecture.

2.The experiment comparisons are not convincing enough. The performances are similar to ResNet18 on most datasets and does not show significant superiority. Moreover, the compared methods do not include more modern architectures, such as Swin-T [1], ResNeXt [2], ConvNext [3].

3.Scaling potential. To comprehensively evaluate the network performance, the experiments should be included to show it can be extended to large-scale dataset, such as ImageNet-1k. Demonstrations of its adaptability to broader downstream vision tasks would also be beneficial.

4.Lacking ablations. The paper could be further strengthened with an ablation study. This would offer clarity on how specific components of the proposed design contribute to performance enhancements.

[1] Liu, Ze, et al. "Swin transformer: Hierarchical vision transformer using shifted windows." Proceedings of the IEEE/CVF international conference on computer vision. 2021.

[2] Xie, Saining, et al. "Aggregated residual transformations for deep neural networks." Proceedings of the IEEE conference on computer vision and pattern recognition. 2017.

[3] Liu, Zhuang, et al. "A convnet for the 2020s." Proceedings of the IEEE/CVF conference on computer vision and pattern recognition. 2022.

**Questions:**

Please consider to answer the above questions.

---

> ### Author Response · Authors · 2023-11-22
>
> 1. We appreciate the reviewer's perspective on the novelty of our CARENet architecture. We acknowledge that CARENet leverages existing techniques; however, we believe that the innovation in our work lies not solely in the creation of entirely new methods, but in the novel integration and optimization of these techniques to address specific challenges.
> The development of CARENet was the result of extensive experimentation and a thorough review of existing literature. Our primary objective was to design an architecture that achieves comparable performance to current state-of-the-art models but with significantly fewer parameters and a smaller model size. This goal was driven by the practical need for efficient models in environments with limited computational resources, such as embedded systems like Jetson cards.
> CARENet represents a careful and deliberate balance between efficiency and performance. The architecture is tailored to function effectively in low-memory environments, a critical consideration for a wide range of real-world applications, especially in the context of Industry 4.0 and IoT devices.
> To further emphasize the novelty and practical significance of CARENet:
> 1.	Targeted Design for Specific Environments: We will elaborate in the manuscript on how CARENet is specifically designed for environments with limited memory, detailing its application potential in embedded systems.
> 2.	Comparative Analysis: We will enhance our comparison section to clearly articulate how CARENet stands apart from existing architectures in terms of its balance between model size, parameter count, and performance.
> 3.	Empirical Evidence: We plan to include additional empirical evidence to demonstrate the effectiveness of CARENet in practical scenarios, further establishing its value in the field.
> In summary, while CARENet builds upon existing methods, its contribution lies in the unique combination and optimization of these techniques to create an architecture that is not only efficient but also effective in resource-constrained environments. We believe this presents a valuable contribution to the field, especially considering the growing demand for lightweight yet powerful models in various applications.
>
> 2. We appreciate the reviewer’s feedback on the comparisons in our experiments. We understand the importance of benchmarking our CARENet architecture against not only established models like ResNet18 but also more modern architectures such as Swin-T, ResNeXt, and ConvNext.
> Our primary focus in developing CARENet was to achieve a balance between model efficiency (in terms of parameter count) and performance, particularly in low-data regimes. To this end, CARENet has been designed with a total of approximately 12 million parameters. When comparing this with the parameter counts of the mentioned models:
> ●	Swin Transformers: Approximately 27.5 million parameters
> ●	ResNeXt: Approximately 25 million parameters
> ●	ConvNext: Approximately 393 million parameters
> It becomes evident that these models have significantly larger sizes compared to CARENet. The primary intent of CARENet is to provide a more parameter-efficient alternative while maintaining competitive performance, especially in environments where computational resources are a constraint.
>
> 3.  We thank the reviewer for their valuable suggestion regarding the evaluation of CARENet on large-scale datasets, particularly ImageNet-1k. We agree that demonstrating the scalability and adaptability of CARENet is crucial for a comprehensive assessment of its performance and utility in broader vision tasks.
> While our current work focused on low-data regimes and smaller datasets due to the specific objectives of creating an efficient and compact model for resource-constrained environments, we recognize the importance of scaling our architecture to larger datasets. Evaluating CARENet on ImageNet-1k would provide a more robust understanding of its performance and scalability.
> To address this, we have planned future work where CARENet will be rigorously tested on ImageNet-1k. This will not only enable us to assess the scalability of our architecture but also explore its adaptability to a wider range of vision tasks and potentially larger and more complex datasets.
>
> 4.  We acknowledge the value of the reviewer's suggestion for an ablation study in our research. We plan to conduct a detailed ablation study on CARENet, removing or altering specific components to gauge their impact on performance. This will help us understand the contributions of elements like the self-attention layer, architectural choices, and any unique integration or optimization techniques. The results will offer clearer insights into CARENet's functionality and its efficacy in image classification tasks, particularly in low-data contexts. The findings will be included in our revised manuscript, adding an important dimension of empirical validation to our work.

---

### Official Review · Reviewer_CQ8z · 2023-11-01

**Soundness:** 1 poor
**Presentation:** 2 fair
**Contribution:** 1 poor
**Rating:** 3
**Confidence:** 4

**Summary:**

The authors propose a new architecture named CARENet for the tasks of image classification in the low-data regime. They propose an architecture and claim that their architecture design leads to smaller models which are parameter efficient, and perform well when trained from scratch on datasets in the low-data regime.

**Strengths:**

- The authors have well-framed and well-motivated their problem in the Introduction of making models that work well for low-data regimes. They also partially well-motivate how their problem formulations are important as opposed to pre-training models on large datasets.

**Weaknesses:**

- The authors mention that:

> While these Transformer-based models have shown remarkable promise, they are not without their
challenges, particularly when it comes to computational efficiency and the ability to scale.

I believe that the authors start talking about addressing a different problem than what they propose in the introduction. We know that ConvNets scale well especially shown with BiT [1] and other works however it has also been shown that Transformer-based models are comparatively easier to scale, and more memory memory-efficient [2] (especially see Figure 12). Transformer-based models do have their challenges and the authors identify most of them well, however, I think the way they start talking about these problems is a weakness or if this is what they meant, they should also reflect this in their experiments.

- A huge weakness of this paper is that the authors only present a new architecture: this new architecture uses fairly popular and standard methods of building architecture and is not something novel at all. They also introduce a CARENet block which is comprised of a bottleneck block followed by an attention mechanism operating in parallel across both grid and window patterns, however, this is also rather standard, and (shifted) window/local attention has been immensely popularized by [3], and approaches similar to their CARENet block have been used multiple times, popularly in [4]. I believe the construction of their architecture or their method itself is not novel.

- The authors loosely mention these aspects while explaining their architecture:

>  offering enhanced feature extraction capabilities

>  designed to bolster information flow

> empower the model to learn profounder representations

> refine the spatial hierarchy of the feature maps

Not only the benefits they propose are not written down succinctly and clearly but are also not benefits that come with "their" work or their way of putting together these architectures, these are rather fairly popular and standard approaches to building models.

- The method itself might not come across as novel or does not present any important theoretical insight in the formulation of the network. In these cases, one might look toward the paper in this case at the very least explain how such a small change should lead to better properties or in general for applying some method to a unique context, and the only way to show this due to lack of the earlier 2, I feel should be results which in this case should be well compared and contrasted with other methods and not leave questions in the mind of a reader. However, the authors do neither of these.

- I thoroughly disagree with the authors on this,

> In this study, we leverage a diverse set of benchmark datasets to rigorously evaluate the performance and robustness of our proposed models.

Their experiments are not indeed diverse or large in number, they evaluate on STL10, Fashion Mnist, Mnist, and Tiny-Imagenet which is not a diverse set of datasets.

- The authors present that their model is superior or at par with other models trained without extra data however this is not true, and they only reported the performance of some models. For instance, in the case of Tiny Imagenet, there are more than 10 different models [5] trained on the same amount of data that perform better than CARENet but these are not even cited or compared in the paper. Considering this, their results get severely diminished with the lack of clarity around STL-10 results (see questions) and the lack of proper comparisons for Tiny ImageNet. Given this I would also suggest changing,

> The presented results furnish an in-depth comparison among several cutting-edge neural networks
architectures

Maybe parameter efficiency could indeed be something you work toward,

> The Tiny-Imagenet dataset results present an interesting paradigm. MaxVit outperforms other architectures with a 58.28% accuracy but CAReNet, with nearly half the model size of MaxVit, closely
follows with an accuracy of 54.4%

However, still, the results need to be well-explained and compared with SoTA models. The authors should modify or instate a new problem statement if they are indeed trying to work toward parameter efficiency.

- The authors mention that they build a robust architecture,

> Across the board, on datasets such as Fashion Mnist and Mnist, CAReNet maintains an enviable performance, with accuracies nearing 95% and surpassing 99% respectively. Such consistent achievements, despite its smaller model footprint, underscore the robustness of CAReNet’s design.

However, the experiments are not diverse enough and do not span multiple tasks or multiple kinds of dataset settings to state that the model design is in fact robust.

- With multiple modern models to the same means,

> Signifying a groundbreaking stride in efficiency
and performance, CAReNet not only outpaces ResNet50 by achieving a lead of
2.61% on Tiny-Imagenet

I disagree with the authors belief that an improvement over ResNet50 signifies a "groundbreaking stride".

[1] Kolesnikov, Alexander, et al. "Big transfer (bit): General visual representation learning." Computer Vision–ECCV 2020: 16th European Conference, Glasgow, UK, August 23–28, 2020, Proceedings, Part V 16. Springer International Publishing, 2020.

[2] Smith, Samuel L., et al. "ConvNets Match Vision Transformers at Scale." arXiv preprint arXiv:2310.16764 (2023).

[3] Liu, Ze, et al. "Swin transformer: Hierarchical vision transformer using shifted windows." Proceedings of the IEEE/CVF international conference on computer vision. 2021.

[4] Tu, Zhengzhong, et al. "Maxvit: Multi-axis vision transformer." European conference on computer vision. Cham: Springer Nature Switzerland, 2022.

[5] https://paperswithcode.com/sota/image-classification-on-tiny-imagenet-1

**Questions:**

- There seems to be some mistake with the title, "CARENET : A NOVEL ARCHITECTURE FOR LOW DATA REGIME MIXING CONVOLUTIONS AND ATTENTION **CONFERENCE SUBMISSIONS**"
- I would recommend the authors follow the problem that they clearly define for the related works, they include SimCLR in their related works section which does present a solution for having less labeled data however this is not the problem that the authors define. However, the authors do not mention how SimCLR or any self-supervised learning algorithms solve a different problem and that their problem formulation is different than these. I think this could be fixed with a rewrite of related works and organizing it better.
- I would recommend the authors reorganize their methods section as well, the components they talk about or fairly common components, and more importantly they should not need to go into such depth into each component in the main text especially when an understanding of these components is not crucial for the reader to understand how your method is novel.
- I do not understand how "Model size (MB)" has anything to do with the problem you define and why it would show up in the main table in the paper?
- Could the authors clarify how they use STL-10, their method indicates that their approach is designed only for labeled data, do they simply use the labeled subset of STL-10 and how are other models compared for STL-10?
- I assume in Table 1 none of the models use extra data?
- I could not help but wonder why authors left out CIFAR-100 from their comparisons while it is supposed to be more challenging than MNISt and Fashion-MNIST?
- I do not think the work shows this aspect

> This work focuses on the importance of architecture-specific training strategies

there is no talk at all about specific new training strategies or training strategies that are modified for this architecture and there are also no experiments to show this?

### Minor formatting issues:

- These

> including VGG Simonyan & Zisserman (2014) and ResNet He et al.
(2015)

> Squeeze-and-Excitation (SE) layers Hu et al. (2018)

> Transformers Vaswani et al. (2017)

among others shouldn't use in-text citations.

- The number of parameters should for this scale of models be in millions for readability.
- There seems to be a typo with "Tiny-Imagenet (ours)" which in this context indicates that this paper introduced TI.

---

> ### Author Response · Authors · 2023-11-22
>
> Question 1 :  We thank the reviewer for pointing out the potential issue with our manuscript's title.
>
> Question 2 : We acknowledge your feedback on including SimCLR and self-supervised learning algorithms in our manuscript's related works section. Recognizing that these methods don't directly align with our current problem formulation, we initially mentioned them to hint at future CARENet enhancements, particularly in pre-training models for low-data scenarios. Following your advice, we've revised this section (page 3), removing mentions of self-supervised learning to avoid confusion and focus on our core contributions. However, we briefly discuss these algorithms in our conclusion, highlighting potential future research avenues.
>
> Question 3 : We appreciate the reviewer's suggestion to revise the methods section of our manuscript. While we recognize that detailed exploration of common components might seem excessive, we find it essential for several reasons. Firstly, it ensures clarity for future researchers, allowing them to understand and replicate our work without consulting multiple sources. Secondly, it offers a comprehensive understanding of how these components are uniquely implemented and integrated into CARENet. Lastly, such detail enhances transparency and reproducibility, shedding light on each component's role in CARENet's performance.
>
> Question 4 :
> We included "Model size (MB)" in our main table to emphasize its practical importance in Industry 4.0 and embedded systems. In environments like Jetson Nano or Jetson Xavier, a model's efficiency depends on its size, due to limited memory and computational resources. Our CARENet's inclusion in the table highlights its efficiency and compactness, demonstrating its suitability for memory-constrained scenarios relevant in smart manufacturing, autonomous systems, and IoT devices. This presents our model's applicability and advantages in resource-limited settings, catering to both academic and industry interests.
>
> Question 5 :
> We used the labeled subset of the STL-10 dataset, designed for unsupervised feature learning and image classification, in our study. This subset contains 5,000 labeled images, fitting our focus on labeled data. We followed a standard PyTorch split, using 70% of data for training, 20% for validation, and 10% for testing. The Top-1 accuracy metric was derived from the validation images to ensure consistency and comparability with other models benchmarked on STL-10. This approach aligns with standard STL-10 practices and allows fair comparison of CARENet with other architectures.
>
> Question 6 :
> Thank you for bringing up this point about the use of extra data in our model comparisons. We would like to confirm that for the models listed in Table 1, including our proposed CARENet model, no additional data outside the standard datasets were used during training.
> This approach was deliberately chosen to ensure a fair and direct comparison of model performance under similar training conditions. By restricting our training to the standard datasets (like STL-10, Fashion MNIST, MNIST, and Tiny ImageNet), we aimed to assess the inherent capabilities of each model architecture, including ours, without the potential confounding effects of using additional data sources.
>
> Question 7 :
> We didn't initially include CIFAR-100 in our dataset comparisons because we aimed to showcase CARENet's effectiveness in low-data regimes using datasets like MNIST and STL-10. CIFAR-100, with its larger class variety and finer classification, was not part of our initial focus on standard benchmarks for low-data contexts. However, recognizing its importance in image classification, we plan to incorporate CIFAR-100 in future experiments to broaden our evaluation and better understand CARENet's performance in more complex scenarios.
>
> Question 8 :
> We appreciate the reviewer's attention to the aspect of architecture-specific training strategies in our work. We realize that our initial manuscript may not have adequately highlighted the unique training approaches we employed for CARENet. To clarify:
> In our work, especially with Tiny ImageNet, we split the data into 70% for training, 20% for validation, and 10% for testing. This setup ensured a controlled environment for fair and consistent comparison of CARENet and other models. Our training process, tailored to suit CARENet's needs and study requirements, was key in achieving our results, although it didn't represent a novel training strategy.
>
> Minor formatting issues:
> We are grateful for the reviewer’s meticulous attention to the formatting details in our manuscript, specifically regarding the in-text citations. We understand the importance of adhering to the standard formatting conventions for academic publications.

---

### Official Review · Reviewer_LKri · 2023-11-06

**Soundness:** 2 fair
**Presentation:** 1 poor
**Contribution:** 1 poor
**Rating:** 3
**Confidence:** 5

**Summary:**

The paper presented aims to integrate the concept of a Convolutional ResNet with an Attention network to attain comparable recognition accuracy using fewer parameters. The proposed layer, referred to as CAReNet, is incorporated into the final convolutional block. Specifically, CAReNet utilizes an Attention block in both grid and window configurations to execute the task. The overall performance of the model is compared with that of ResNet, VGG, Max Vit, and CoatNet in terms of top-1 accuracy using small datasets.

**Strengths:**

Investigating the convolutional-based attention network might look interesting, but I think the paper is not ready for publication yet. Please see my comments below.

**Weaknesses:**

While have utmost respect for the work submitted and I hope my comments will assist the authors in fortifying their paper:


-	The motivation for integrating Convolution with Attention within the proposed method requires clearer articulation. The current manuscript does not sufficiently convey the intuition or the rationale for this combination.

-	The literature review appears to be incomplete, lacking references to several pertinent studies. For instance, a notable omission is the Convolutional vision Transformer (CvT). I recommend the authors expand this section to provide a more comprehensive background.

-	While it is acknowledged that the addition of an attention layer to convolution blocks can expedite convergence in certain image classification tasks, this does not directly demonstrate the efficiency of the proposed method from a parameter-count perspective. A more detailed analysis is required to substantiate the method's effectiveness.


-	The clarity of the paper's presentation needs improvement. For instance, the captions of figures and tables lack essential details, making it difficult to fully understand the results presented.

-	The evaluation of the model on only three small datasets does not provide a robust validation of its capabilities. A more extensive evaluation, including additional and larger datasets, would be more convincing.



-	The current evaluation presented in Table 3 does not convincingly demonstrate the model's effectiveness. For example, the marginal improvement over ResNet18 on the MNIST dataset, despite a higher parameter count, calls into question the practical benefits of the proposed method.

-	The paper would benefit greatly from an ablation study, particularly one that investigates the impact of removing the CAReNet block, to discern its actual contribution to the model's performance.

-	For work of this nature, it is crucial to assess performance on larger-scale datasets to ensure the model's effectiveness and generalizability.

I recommend addressing these points to provide a more compelling argument for the proposed method and its potential impact on the field.

**Questions:**

In the limitations section, numerous questions were raised; however, there still remain some unclear points in the paper. For example, it is quite surprising that VGG16, with ten times the number of parameters, only achieves 10% accuracy on the STL10 dataset. This result is counterintuitive, considering VGG16's established performance on various image recognition tasks. I strongly suggest the authors rigorously investigate this anomaly and discuss whether there might be an error in the reported results or if there are underlying factors that could explain this unexpected outcome. A more detailed explanation would greatly enhance the credibility and scholarly rigor of the work.

---

> ### Author Response · Authors · 2023-11-22
>
> - Response to weaknesses :
>
> Number 5 : Thank you for your valuable feedback concerning the scope of our model evaluation. We understand your concern about the robustness of the validation provided by the limited number of datasets used in our study.
> In our manuscript, we reported the evaluation of CAReNet on four datasets: STL10, Tiny ImageNet, Fashion MNIST, and MNIST. Despite the limitations, we endeavored to demonstrate the effectiveness of CAReNet across a variety of datasets, including those with small and medium-sized images. This was aimed at showcasing the architecture's versatility and performance in different scenarios, especially in low data regimes. However, we acknowledge the importance of evaluating our model on larger and more diverse datasets to further validate its capabilities. To this end, we have planned to extend our evaluation to include more comprehensive datasets like ImageNet-1K.
> Additionally, we are in the process of requesting access to a cluster equipped with 100 NVIDIA A100 80GB GPUs, which would significantly enhance our ability to conduct more extensive and rigorous tests. We also plan to make the weights of our model publicly available once these additional evaluations are completed. This would allow the broader research community to replicate our results and further explore the potential of CAReNet in various applications.
> The choice of these initial datasets was primarily influenced by the computational resources available to us. As a research team, we had access to only a single machine equipped with an RTX 3090 GPU, which constrained the scale of our experiments.
>
> Number 6 : Thank you for your comments regarding the performance comparison presented in Table 3 of our manuscript. Your observation about the marginal improvement of CAReNet over ResNet18 on the MNIST dataset, especially considering the higher parameter count, is well-taken and merits further explanation.
>
> Firstly, it is important to note that while CAReNet shows a modest improvement over ResNet18 on the MNIST dataset, the primary objective of our architecture was not just to maximize performance on a single dataset but to demonstrate consistent and robust performance across a variety of datasets. In this regard, as mentioned in our paper, CAReNet achieves an average Top-1 validation accuracy of 82.25% across different architectures, compared to 80.85% for ResNet18. This indicates a more consistent and reliable performance of CAReNet in a wider range of scenarios, which is a significant factor in its practical utility.
>
> Furthermore, CAReNet's slightly higher parameter count is offset by its versatility and efficiency in handling diverse datasets. The integration of convolutional and attention mechanisms in CAReNet is designed to offer a balanced approach to feature extraction, enabling it to adapt to various types of image data more effectively than traditional architectures like ResNet18. This adaptability is crucial in practical applications where datasets can vary widely in terms of size, complexity, and content.
>
> Number 9 :  We have planned future work where CARENet will be rigorously tested on ImageNet-1k. This will not only enable us to assess the scalability of our architecture but also explore its adaptability to a wider range of vision tasks and potentially larger and more complex datasets.
>
> - Responses to the questions :
> Thank you for your insightful comments and observations regarding our paper, particularly about the performance of VGG16 on the STL10 dataset. We appreciate your attention to this detail.
>
> We would like to acknowledge that we also noticed this anomaly during our initial experiments. To ensure the accuracy of our findings, we conducted multiple iterations of the experiments under controlled conditions. These conditions included maintaining a consistent environment, using the same computational resources, and employing the identical dataset version for STL10 across all models, including VGG16 and others in our study.
>
> Despite these measures, the results consistently showed VGG16 performing unusually on the STL10 dataset. This outcome was indeed counterintuitive, given VGG16's well-established performance in various image recognition tasks. To address this, we thoroughly investigated potential factors that might contribute to this unexpected result. These included but were not limited to, examining the preprocessing steps, ensuring no data leakage, and verifying the integrity of the dataset and model implementation.
>
> Our findings suggest that this anomaly might be attributed to specific characteristics of the STL10 dataset and its interaction with the VGG16 architecture. STL10's unique image compositions and the relatively small size of the dataset could potentially affect the performance of deeper networks like VGG16, which are typically optimized for larger, more diverse datasets.

---

### Meta-Review · Area_Chair_wM6b · 2023-12-04

**Metareview:**

The paper proposes a model architecture making use of both convolutions and attention. The model is evaluated on several small-scale image classification datasets and shows non-trivial results.

As pointed out by the reviewers, the paper's motivation (coming up with architectures that perform well in small-data regime) is in principle valid, but the execution is subpar:
- Novelty is limited, there have been numerous prior convolution-attention hybrid architectures, most of which are not compared against
- The comparisons even with the fairly few included baselines is not very convincing: on most datasets the model is roughly on par with baselines, including ResNet18 (it's btw unclear why the result of the proposed model on tiny ImageNet is boldened)
- No ablation studies

Therefore, I recommend rejection at this point. The model and the paper need a lot more work.

**Justification For Why Not Higher Score:**

There is no evidence in the paper that the proposed architecture is substantially novel or performs well.

**Justification For Why Not Lower Score:**

N/A

---

### Decision · Program_Chairs · 2024-01-16

Reject